# Sex Differences in Proatherogenic Cytokine Levels

**DOI:** 10.3390/ijms21113861

**Published:** 2020-05-29

**Authors:** Stella Bernardi, Barbara Toffoli, Federica Tonon, Morena Francica, Elena Campagnolo, Tommaso Ferretti, Sarah Comar, Fabiola Giudici, Elisabetta Stenner, Bruno Fabris

**Affiliations:** 1Department of Medical, Surgical, and Health Sciences, University of Trieste, Cattinara Teaching Hospital UCO Medicina Clinica, 34100 Trieste, Italy; effe.tonon@gmail.com (F.T.); morefrancica@gmail.com (M.F.); campagnoloelena@gmail.com (E.C.); tomferretti@yahoo.com (T.F.); sarah.comar@studenti.units.it (S.C.); fgiudici@units.it (F.G.); b.fabris@fmc.units.it (B.F.); 2ASUGI Azienda Sanitaria Universitaria Integrata di Trieste, Cattinara Teaching Hospital, UCO Medicina Clinica, 34100 Trieste, Italy; 3Institute for Maternal and Child Health, IRCCS Burlo Garofolo, 34100 Trieste, Italy; barbaratoffoli.ts@gmail.com; 4Department of Life Sciences, University of Trieste, 34100 Trieste, Italy; 5Unit of Biostatistics, Epidemiology and Public Health, Department of Cardiac, Thoracic, Vascular Sciences and Public Health, University of Padua, 35100 Padova, Italy; 6Department of Diagnostics, Azienda USL Toscana Nordovest, 57100 Livorno, Italy; elisabetta.stenner@uslnordovest.toscana.it

**Keywords:** sex, gender, cytokines, inflammation, atherosclerosis, IL-1β, IL-6, TNF-α, OPG, ACE2, healthy adults

## Abstract

Background: It has been shown that sex affects immunity, including cytokine production. Given that atherosclerosis is an inflammatory disease promoted by specific cytokines, such as interleukin (IL)-1β, IL-6, and tumor necrosis factor (TNF)-α, we aimed at evaluating whether sex could affect the levels of these proatherogenic cytokines in a group of healthy adults. In this analysis, we also included other cytokines and peptides that have been implicated in atherosclerosis development and progression. Methods: A total of 104 healthy adults were recruited; we measured circulating levels of IL-1β, IL-6, TNF-α, angiotensins and angiotensin-converting enzyme-2 (ACE2), as well as osteoprotegerin and receptor activator of nuclear factor κB ligand (RANKL). Results: IL-1β, IL-6, and TNF-α were significantly higher in men as compared to women. They were all associated with testosterone and the testosterone/estradiol ratio. They remained significantly associated with sex (but not with hormones) after being tested for potential confounders. Conclusions: Sex seems to influence the levels of proatherogenic cytokines. This is consistent not only with sex differences in vulnerability to infections but also with the higher cardiovascular risk exhibited by the male gender as compared to the female gender. Nevertheless, this association is only partly explained by hormone levels.

## 1. Introduction

It has been shown that sex affects immunity [1]. Women have a higher incidence of autoimmune diseases [2] and a lower burden of microbial infections as compared to men [1], as they exhibit higher immunoglobulin levels [3] and stronger innate and adaptive immune responses than men [4]. Recently, for example, it has emerged that there are also sex differences in mortality and vulnerability to Covid-19 infection [5], as the severity of disease and death is greater in men [6]. It has been reported that the sex differences in immune responses might include cytokine production, which are low-molecular-weight-proteins that mediate the immune responses between cells [7]. In particular, if women display greater type 2 cytokine production, then men seem to display greater type 1 cytokine production [8,9], which include interleukin (IL)-1β and tumor necrosis factor (TNF)-α.

Atherosclerosis is a chronic inflammatory disease mediated by innate and adaptive immunity [10]. In this setting, cytokines, which enable the crosstalk between immune and nonimmune cells, are involved in endothelial dysfunction, leukocyte migration, plaque formation and development, as well as its eventual rupture [11]. The cytokines contributing to atherosclerosis can be broadly divided into proatherogenic (such as IL-1β, IL-6, TNF-α) and antiatherogenic (such as IL-10 and IL-1rA). Recently, two clinical trials demonstrated that targeted cytokine inhibition, rather than broad-spectrum anti-inflammatory therapy, can significantly reduce adverse cardiovascular events [12,13]. In addition, it has been argued that an imbalance between type 1 and type 2 cytokine production may promote coronary artery disease [11,14]. As such, any difference in cytokine production between men and women could favor plaque development, leading to a higher risk of cardiovascular events.

In order to evaluate whether sex could affect circulating proatherogenic cytokines, we measured IL-1β, IL-6, and TNF-α in a group of healthy adults. In addition to these cytokines, we looked at other cytokines and circulating mediators that are associated with proatherogenic cytokines [15] and that are involved in tissue inflammation and atherosclerosis. These cascades include the renin–angiotensin system [16,17] as well as the system of osteoprotegerin (OPG) and its ligand-receptor activator of NF-κB (RANKL) [18,19].

## 2. Results

### 2.1. General Characteristics of the Population

We recruited 104 healthy adults (68 women (65%) and 36 men (35%)), whose general characteristics are reported in Table 1. All the subjects were Caucasian. Groups did not differ in terms of age. All the subjects either worked or studied in the same hospital, as the subjects recruited were 76 medical school students and 28 physicians. The students were 52 women (68%) and 24 men (32%), while the physicians were 16 women (57%) and 12 men (43%). It has to be noted that all these subjects undergo periodically routine medical check-ups by the occupational health service, and none of them had evidence of disease. In both groups, general characteristics were within reference ranges (Table 1). Nevertheless, women exhibited lower body mass index (BMI), lower systolic blood pressure (SBP) and diastolic blood pressure (DBP), and higher high density lipoprotein (HDL) and total cholesterol values. Levels of estradiol and testosterone differed according to biological sex.

### 2.2. Cytokine Levels

Circulating levels of IL-1β, IL-6, and TNF-α were not always detectable and their distribution was not normal. Median IL-1β was 0 pg/mL (0–386.4). Median IL-6 was 17.2 pg/mL (0–600). Median TNF-α was 0 pg/mL (0–1000). Circulating levels of angiotensin-converting enzyme-2 (ACE2), angiotensin (Ang) II, Ang 1-7, OPG, and RANKL were always detectable and their distribution was not normal. Median ACE2 was 61.92 ng/mL (33.69–246.66). Median Ang II was 388.2 pg/mL (54.23–2970.8). Median Ang 1-7 was 212.9 pg/mL (56.1–2227.9). Median Ang II/Ang 1-7 ratio was 1.7 (0.3–5.2). Median OPG was 697.4 pg/mL (382.2–1773.8). Median RANKL was 38.9 pg/mL (0–824). Circulating levels of IL-1β, IL-6, OPG, TNF-α, as well as Ang II and Ang 1-7 in women and men are reported in Figure 1. Men exhibited significantly higher levels of IL-1β, IL-6, and TNF-α (*p* = 0.010, *p* = 0.019, and *p* = 0.024), while women exhibited significantly higher levels of OPG (*p* = 0.035). ACE2, Ang II, Ang 1-7, as well as the Ang II/Ang 1-7 ratio, and RANKL did not differ between the groups.

### 2.3. Linear Correlations between Variables

Given that ACE2, Ang II, Ang 1-7, and RANKL did not differ between women and men, we looked at the correlations between the levels of IL-1β, IL-6, TNF-α, OPG, and baseline general and biochemical variables. All proinflammatory cytokines (IL-1β, IL-6, TNF-α) were significantly associated with circulating testosterone and testosterone/estradiol (T/E2) ratio (Table 2) but not with estradiol. In addition, IL-1β was also significantly associated with age and glucose. When we analyzed the group of women separately from that of men, the associations between testosterone and proinflammatory cytokines were no longer significant (Table 3). On the other hand, OPG was associated not only with testosterone (inversely) but also with age (directly), as well as systolic and diastolic blood pressure (inversely), as shown in Table 4. When we analyzed the group of women separately from that of men, these associations remained significant in the subgroup of women (Table 4, Figure 2).

### 2.4. Multivariate Analysis

The multivariate regression model showed that sex was independently associated with IL-6 (*p* = 0.024) and TNF-α (*p* = 0.031), but not with IL-1β and OPG (Table 5). We did not test sex and testosterone together because of collinearity. The multivariate regression model was repeated to test the association of testosterone and T/E2 with proatherogenic cytokines, but hormones were not independently associated with any of them. IL-6, TNF-α, and OPG were also independently associated with age (Table 5). In the subgroup of women, OPG remained independently associated with age (*p* < 0.01) and with estradiol (*p* < 0.01), but not with blood pressure levels.

## 3. Discussion

This work demonstrates that healthy men have higher levels of circulating IL-1β, IL-6, TNF-α as compared to women. Our finding is in line with studies that demonstrate that there is a sex difference in cytokine production in humans. For instance, in a study that included more than 500 healthy subjects (blood donors), it was found that the production of proinflammatory cytokines (IL-1β, IL-6, and TNF-α) released from monocytes after stimulation was higher in men, while women exhibited a higher production of lymphocyte-derived cytokines (IL-7 and IL-22) [7]. Other investigators also reported that men had a higher production of monocyte-derived IL-1β, IL-6, and tumor necrosis factor (TNF)-α upon stimulation [9,20] and a decreased percentage of lymphocytes producing IL-2 as compared to women [20]. The stronger monocyte-derived cytokine production in men has been recently confirmed in a pooled analysis of 15 study populations, where the investigators showed that it was present at all ages regardless of specific diseases, such as osteoarthritis, rheumatoid arthritis, multiple sclerosis, systemic lupus erythematosus, or cardiac diseases [21].

Estrogens, progesterone, and testosterone significantly influence innate and adaptive immunity, as reviewed by Klein and Flanagan [4]. Nevertheless, studies on the effects of gonadal hormones on cytokine production report conflicting results [21,22], and the magnitude of hormone effects is debatable [23]. In our study, although we found a significant—albeit weak—correlation between testosterone and proinflammatory cytokine levels, multiple logistic models showed that sex was independently associated with cytokine levels, but not with testosterone or testosterone/estradiol ratio, indicating that hormone levels do not fully explain sex differences in immune responses. Our data are consistent with the observations of Ter Horst et al., who assessed the correlation of inflammatory markers with the levels of progesterone and testosterone in men and women, finding that the majority of cytokines showed no correlation with progesterone and testosterone, ruling out a major role of hormones in explaining the sex difference [7]. In addition, our data are consistent with the current theory of sexual differentiation of all mammalian tissues, whereby the gonadal hormones are relevant to sex differences in phenotype as much as other secondary factors that are downstream of the primary sex-specific effects of X and Y genes [24].

It has been argued that the higher monocyte-derived cytokine production in men may contribute to the increased susceptibility of men to atherosclerosis and cardiovascular disease [25]. IL-1β, IL-6, and TNF-α are soluble low-molecular-weight-proteins that mediate crosstalk between immune and nonimmune cells and have been implicated in atherosclerosis development and progression [26]. IL-1β belongs to the IL-1 cytokine family, whose members are expressed in atherosclerotic plaques [11]. Animal studies have shown that blockade of IL-1 signaling with IL-1 receptor antagonist diminishes the size of atherosclerotic lesions [27,28]. Most importantly, it has been recently demonstrated that patients with previous myocardial infarction, who received the monoclonal antibody targeting IL-1β selectively, had a 15% reduction of major adverse cardiovascular events (MACE) as well as a 17% reduction of MACE and hospitalization for unstable angina requiring urgent revascularization after 48 months of treatment [12]. Likewise, IL-6 is also expressed in human atherosclerotic plaques. Several epidemiological studies have highlighted the association between IL-6 signaling and cardiovascular disease onset and complications. In apparently healthy men, elevated levels of IL-6 were associated with increased risk of future myocardial infarction [29]. In addition, in patients with unstable coronary artery disease, circulating IL-6 was a strong independent marker of increased mortality [30]. More recently, in a collaborative meta-analysis, the presence of a minor genetic variant that impairs IL-6 signaling was associated with a reduction of coronary artery disease, indicating the presence of a causal association between IL-6 receptor-related pathways and coronary artery disease [31]. As for TNF-α, earlier experimental studies showed a similar involvement in plaque development and progression [11].

In addition to IL-1β, IL-6, and TNF-α, other circulating peptides and cytokines are linked to atherosclerosis and cardiovascular disease. For instance, it has been clearly demonstrated that the renin–angiotensin system (RAS) and the angiotensins, which are its mediators, promote atherosclerosis development and progression through local inflammation [31]. For this reason, we measured Ang II and Ang 1-7, which are considered the final effectors of the RAS, as well as ACE2, which is considered the key regulator of Ang II and Ang 1-7 levels. In this study, we did not find any difference in Ang II, Ang 1-7, the Ang I /Ang 1-7 ratio, as well as in ACE2 between men and women. Nevertheless, circulating levels might not reflect tissue levels of enzymes and peptides [32,33], which could explain our negative result. More recently, a glycoprotein belonging to the TNF superfamily, which is OPG, has been linked to tissue inflammation, atherosclerosis, and cardiovascular disease. Over the last decades, OPG has been associated with the risk of future coronary artery disease in apparently healthy men and women, independent of established cardiovascular risk factors [18,32]. Moreover, in patients with acute coronary syndromes, OPG has been linked to the incidence of death, hospitalizations, myocardial infarction, and stroke [33], which has been successively observed in the general population as well [34]. Nevertheless, it remains unclear if OPG is a risk marker or a risk factor of atherosclerosis and cardiovascular disease [19,35]. In this study, we measured OPG and its ligand RANKL, and found that women had higher levels of OPG. This finding is consistent with the literature, reporting that premenopausal women had higher OPG levels as compared to men under the age of 50 years [36], which should be ascribed to the stimulatory effect that estrogen has on OPG expression in osteoblast cells [37], where it inhibits bone resorption. Nevertheless, when the association between OPG and sex was tested for other potential confounders, it was no longer found significant. In the subgroup of women alone, OPG turned out to be independently associated with age (directly) and estradiol (indirectly), which is consistent with the findings of other investigators [38,39].

It has been argued that the stronger monocyte-derived cytokine production in men may contribute not only to the higher cardiovascular risk, but also to the higher vulnerability to infections, as compared to women. It has been shown that higher IL-6 and TNF-α levels are associated with a higher risk of developing sepsis [40], and that, among patients with sepsis, men had worse outcomes, higher TNFα, and lower IL-10 levels than women [41]. On March 11, 2020, the World Health Organization declared the outbreak of the severe acute respiratory syndrome-coronavirus (SARS-CoV2) a global pandemic. Early epidemiological observations indicate that the severity of disease and death is two times greater for men than women [6]. Some investigators have ascribed such a different response to IL-6 levels, which is linked to poor survival in patients with acute respiratory distress syndrome [42], and which is lower in women after viral infection [43]. Other explanations include a different expression of ACE2 [44], which has been recently identified as the SARS-CoV-2 receptor [45]. In our study, we found a sex difference in circulating IL-6 but not in ACE2, which may be due to the fact that ACE2 should be measured in tissues rather than in the circulation [45].

In conclusion, in a cohort of healthy adults, sex seems to influence the levels of proinflammatory/proatherogenic cytokines, as we found that men had higher levels of IL-1β, IL-6, and TNF-α as compared to women. This is consistent not only with sex differences in vulnerability to infections but also with the higher cardiovascular risk exhibited by men as compared to premenopausal women, but it is only partly explained by different hormone levels.

## 4. Materials and Methods

### 4.1. Population

Between January 2019 and December 2019, we recruited 104 adults (68 women and 36 men). All the individuals were recruited among medical school students and physicians working at Cattinara Teaching Hospital (ASUITS). It has to be noted that all these subjects undergo periodically routine medical check-ups by the occupational health service (including medical visits, laboratory tests, and EKGs, if appropriate) with no evidence of disease. Inclusion criteria were a history of normal sexual development for adults. Exclusion criteria were the use of medication, the onset of a fever, cold, or coryza within 14 days of blood sampling, the presence of any known disease, the cessation of menses (for adult women), and an unwillingness to participate in this study. All subjects underwent a medical visit in order to record their medical history as well as general characteristics (such as blood pressure, height, and weight). Blood sampling was performed at fasting, and blood samples were collected in two vacutainer tubes (2 × 10 mL for adults); one tube containing a clot activator and a gel separator to get the serum, the other containing ethylenediaminetetraacetic acid (EDTA) to get the plasma. The study was approved by our regional ethics committee on the 27/09/2019 (CEUR-2019-SPER-113). Inclusion of volunteers and experiments were conducted in accordance with the principles expressed in the Declaration of Helsinki. All subjects gave their written informed consent before participation in the study.

### 4.2. Biochemical Characteristics and Cytokine Measurement

Glucose, triglycerides, total cholesterol, and HDL cholesterol were measured by autoanalyzer. LDL cholesterol was calculated by the Friedwald’s formula. Estradiol (#07027249190, Roche, Basel, Switzerland) and testosterone (#07027915190, Roche, Basel, Switzerland) were measured by electrochemiluminescence immunoassay on a Roche cobas e 801 system. Testosterone (nmol/L)/Estradiol (pmol/L) ratio was measured with the following formula: testosterone/(10* Estradiol). Interleukin-1beta (IL-1β), interleukin-6 (IL-6), tumor necrosis factor (TNF)-α, osteoprotegerin (OPG), receptor activator of nuclear factor κB ligand (RANKL), angiotensin (Ang)II, Ang 1-7, and angiotensin-converting enzyme 2 (ACE2) were measured by ELISA. IL-1β was measured with the R&D #DY201 ELISA kit (R&D Systems, Minneapolis, Minnesota, United States); IL-6 was measured with the R&D #DY206 ELISA kit (R&D Systems, Minneapolis, Minnesota, United States); TNF-α was measured with the R&D #DY210 ELISA kit (R&D Systems, Minneapolis, Minnesota, United States); Ang II was measured with the Elabscience #E-EL-H0326 ELISA kit (Elabscience, Houston, Texas, United States); Ang 1-7 was measured with the Elabscience #E-EL-H5518 ELISA kit (Elabscience, Houston, Texas, United States); ACE2 was measured with the Elabscience #E-EL-H0281 ELISA kit (Elabscience, Houston, Texas, United States); OPG was measured with the R&D #DY805 ELISA kit (R&D Systems, Minneapolis, Minnesota, United States); RANKL was measured with the R&D #DY626 ELISA kit (R&D Systems, Minneapolis, Minnesota, United States). The ELISA kits were evaluated for intra-assay reproducibility by running 3 positive control samples (containing high, medium, and low concentration of the specific marker) in duplicate (coefficient of variation (CV %) was <10%). For the inter-assay reproducibility, three control samples of known concentration were tested in duplicate in separate plates and on different days (CV < 15%).

### 4.3. Statistics

Continuous variables were checked for normal distribution by visual inspection of the normal probability plot and performing the Shapiro–Wilk normality test. Continuous variables were reported as median with range (minimum–maximum). Categorical variables were reported as percentages. Continuous variables were compared by the non-parametric Mann–Whitney test for independent data. Categorical variables were compared by the Chi-Square test or Fischer’s exact test, whenever appropriate. Linear correlations between variables of interest were analyzed with the Spearman coefficient [46]. A multiple linear regression model was used to evaluate if the association between cytokines and sex was independent of other possible confounders. Sex and testosterone were not evaluated together for collinearity. Statistical analyses were conducted with the software R (version 3.5.3) and *p*-values less than 5% were considered as evidence of a significant association.

## Figures and Tables

**Figure 1 ijms-21-03861-f001:**
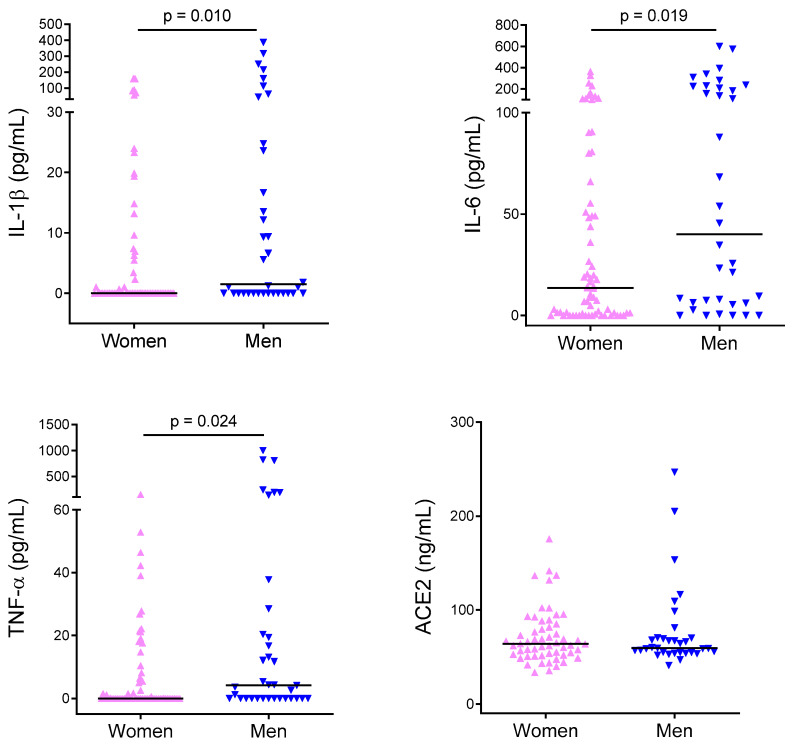
Proatherogenic cytokine levels in women and men. Circulating levels of IL-1β, IL-6, TNF-α, as well as angiotensin-converting enzyme-2 (ACE2), Ang II, Ang 1-7, osteoprotegerin (OPG), and receptor activator of nuclear factor κB ligand (RANKL). Bars are median values.

**Figure 2 ijms-21-03861-f002:**
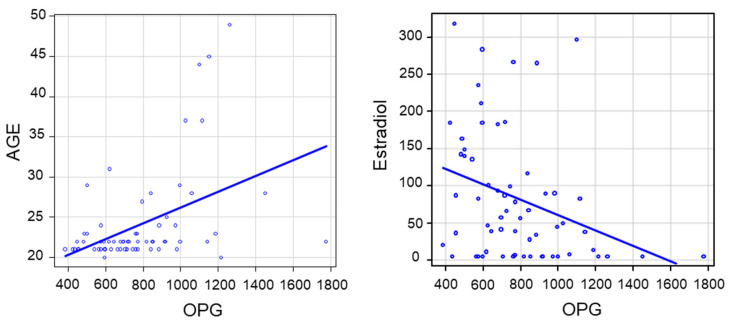
Correlations between age and OPG, and estradiol and OPG, in the subgroup of women.

**Table 1 ijms-21-03861-t001:** General characteristics of women and men.

	Women (*n* = 68)	Men (*n* = 36)	*p*-Value
Age (years)	22 (20–49)	22 (21–37)	0.997
Smoke (%)	21%	15%	0.49
CVD Family history (%)	39%	33%	0.59
Physical activity (%)	72%	70%	0.84
BMI (weight/height^2^)	21 (17–36)	23.2 (19.8–31.4)	<0.001
SBP (mmHg)	115 (90–140)	125 (100–145)	<0.001
DBP (mmHg)	70 (60–90)	80 (60–90)	<0.01
Glucose (mg/dL)	85 (65–100)	86 (74–123)	0.06
Total Cholesterol (mg/dL)	185 (129–254)	159 (95–249)	<0.05
HDL-C (mg/dL)	61 (38–94)	46 (33–64)	<0.001
Triglycerides (mg/dL)	66 (37–136)	68 (40–271)	0.4
LDL-C (mg/dL)	111 (58–169)	97 (39–179)	0.14
Estradiol (pg/mL)	53 (7.8–319)	23 (9–45)	<0.05
Testosterone (ng/mL)	0.3 (0.02–0.6)	5.7 (2.8–8.2)	<0.001

Values are reported as median (min–max). BMI is for body mass index, SBP is for systolic blood pressure, and DBP is for diastolic blood pressure. HDL-C is for high density lipoprotein cholesterol and LDL-C is for low density lipoprotein cholesterol.

**Table 2 ijms-21-03861-t002:** Correlation between proinflammatory cytokines and clinical/hormonal variables in the entire cohort.

Variable	IL1-β		IL-6		TNF-α	
	ρ	*p*-Value	ρ	*p*-Value	ρ	*p*-Value
Age	0.23	0.02	0.14	0.16	0.09	0.35
BMI (weight/height^2^)	0.15	0.12	0.13	0.21	0.19	0.07
SBP (mmHg)	0.17	0.09	0.17	0.09	0.12	0.25
DBP (mmHg)	0.14	0.17	0.15	0.13	0.09	0.36
Glucose	0.26	0.008	0.14	0.17	0.18	0.08
Total cholesterol	−0.10	0.30	−0.19	0.06	−0.15	0.13
HDL cholesterol	−0.05	0.63	−0.18	0.13	−0.04	0.65
LDL cholesterol	−0.12	0.22	−0.15	0.13	−0.18	0.08
Triglycerides	−0.01	0.89	−0.03	0.79	−0.09	0.35
Estradiol	−0.09	0.38	−0.09	0.35	−0.14	0.19
Testosterone	0.25	0.01	0.20	0.04	0.18	0.08
T/E2	0.28	0.005	0.24	0.02	0.27	0.007

**Table 3 ijms-21-03861-t003:** Correlation between proinflammatory cytokines and clinical/hormonal variables in the subgroups of women and men.

	Women					
Variable	IL1-β		IL-6		TNF-α	
	ρ	*p*-Value	ρ	*p*-Value	ρ	*p*-Value
Age	0.11	0.37	0.04	0.74	0.04	0.75
BMI (weight/height^2^)	0.09	0.46	0.13	0.29	0.13	0.29
SBP (mmHg)	0.06	0.64	0.02	0.88	−0.02	0.86
DBP (mmHg)	0.04	0.72	0.02	0.87	−0.03	0.82
Glucose	0.19	0.13	0.01	0.99	0.10	0.44
Total cholesterol	−0.02	0.88	−0.05	0.71	−0.11	0.39
HDL cholesterol	0.09	0.44	−0.07	0.54	0.07	0.56
LDL cholesterol	−0.09	0.48	−0.04	0.75	−0.18	0.14
Triglycerides	0.08	0.52	0.14	0.26	−0.03	0.82
Estradiol	−0.01	0.91	−0.07	0.57	−0.10	0.45
Testosterone	0.01	0.99	−0.05	0.72	−0.10	0.42
E2/T	−0.06	0.64	−0.08	0.51	−0.09	0.49
T/E2	0.02	0.87	0.05	0.67	0.05	0.68
	**Men**					
**Variable**	**IL1-β**		**IL-6**		**TNF-α**	
	**ρ**	***p*-Value**	**ρ**	***p*-Value**	**ρ**	***p*-Value**
Age	0.45	0.007	0.34	0.05	0.16	0.36
BMI	0.04	0.82	−0.14	0.46	0.02	0.90
SBP	0.04	0.80	0.17	0.35	0.05	0.79
DBP	0.11	0.53	0.18	0.32	0.11	0.56
Glucose	0.28	0.11	0.31	0.07	0.19	0.29
Total cholesterol	−0.07	0.67	−0.23	0.19	−0.05	0.77
HDL cholesterol	0.10	0.57	0.08	0.67	0.18	0.30
LDL cholesterol	−0.08	0.64	−0.20	0.25	−0.07	0.69
Triglycerides	−0.16	0.36	−0.31	0.07	−0.25	0.15
Estradiol	−0.09	0.58	0.03	0.85	−0.16	0.37
Testosterone	−0.02	0.89	0.19	0.27	0.02	0.89
T/E2	0.09	0.60	0.09	0.61	0.21	0.23

**Table 4 ijms-21-03861-t004:** Correlation between OPG and clinical/hormonal variables in the entire cohort and the subgroups of women and men.

Variable	Entire Cohort		Women		Men	
	ρ	*p*-Value	ρ	*p*-Value	ρ	*p*-Value
Age	0.35	0.0004	0.42	0.0004	0.24	0.17
BMI	−0.21	0.03	−0.12	0.31	−0.19	0.30
SBP	−0.22	0.03	−0.26	0.03	0.01	0.94
DBP	−0.32	0.001	−0.30	0.01	−0.33	0.06
Glucose	0.06	0.53	0.03	0.79	0.22	0.21
Total cholesterol	0.03	0.77	0.08	0.52	−0.23	0.19
HDL cholesterol	0.15	0.14	0.08	0.50	0.03	0.85
LDL cholesterol	−0.04	0.72	−0.01	0.97	−0.19	0.26
Triglycerides	−0.02	0.80	0.13	0.30	−0.28	0.11
Estradiol	−0.21	0.04	−0.36	0.004	0.008	0.96
Testosterone	−0.31	0.001	−0.37	0.002	0.06	0.72
T/E2	−0.05	0.58	−0.01	0.91	0.09	0.59

**Table 5 ijms-21-03861-t005:** Association of sex with proatherogenic cytokines in the entire patient cohort.

Dependent Variable IL-1β			
Predictive Variables	β-Estimate	Standard Error	*p*-Value
Sex	27.14	13.52	0.09
Age	2.75	1.24	0.03
BMI	0.27	1.91	0.88
SBP	0.15	0.54	0.78
Glucose	1.27	0.78	0.11
**Dependent Variable IL-6**			
Predictive Variables	β-Estimate	Standard Error	*p*-Value
Sex	61.41	26.88	0.02
Age	5.61	2.51	0.03
BMI	0.46	3.89	0.90
SBP	0.69	1.11	0.53
Glucose	2.18	1.67	1.94
**Dependent Variable TNF-α**			
Predictive Variables	β-Estimate	Standard Error	*p*-Value
Sex	70.39	32.11	0.03
Age	8.91	3.24	0.007
BMI	−0.04	4.65	0.99
SBP (mmHg)	0.14	1.35	0.91
Glucose (mg/dL)	0.78	1.99	0.69
**Dependent Variable OPG**			
Predictive Variables	β-Estimate	Standard Error	*p*-Value
Sex	−75.07	54.96	0.17
Age	14.77	5.55	0.009
BMI	−7.10	7.96	0.37
SBP	−2.22	2.31	0.34
Glucose	2.75	3.41	0.42

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
