# Peer review of "Sex Differences in Proatherogenic Cytokine Levels"

_ijms, 2020, doi:10.3390/ijms21113861_

Round 1

Reviewer 1 Report

Gender difference in proatherogenic cytokine levels

This is an interesting paper. Some comments:

  1. Have you considered any transformations of the proteins, especially those that were not normally distributed?
  2. For pairs of continuous variables have their relationships been assessed for linearity? It would be interesting to see plots of one variable against the other, such as scatterplots.
  3. Section 2.3 “When we analyzed separately the group of women from that of men, the correlation between testosterone and proinflammatory cytokines could no longer be found in any group.” Does this statement mean that the associations were no longer significant or that the direction or magnitude of the associations changed? Please clarify.
  4. It would be good to add analysis as in table 2 separately by sex. Even if associations are not significant it is useful to assess whether there is any heterogeneity in associations between males and females.
  5. Have the assays used been assessed for reproducibility?
  6. I suggest to use the word ‘parameters’ only for statistical parameters to be estimated and use ‘variables’ or something else for quantities that are measured.
  7. I suggest instead of ‘the significant level was set at 5%’ to say ‘p-values less than 5% were considered as evidence of an association’ or something similar.

Author Response

We thank the Reviewer 1 for the valuable feedback which has helped us improve the quality of the manuscript. Please find below our point-by-point reply. The changes have been highlighted by blue ink throughout the text.

1. Have you considered any transformations of the proteins, especially those that were not normally distributed?

Reply: We ascribed the distribution of the proteins to the distribution of age and sex which were bimodal, as proteins were correlated to both age and se

2. For pairs of continuous variables have their relationships been assessed for linearity? It would be interesting to see plots of one variable against the other, such as scatterplots.

Reply: After evaluating all the associations between continuos variables, we decided to report the correlations with the best rho values, which were the correlations between AGE and OPG and estradiol and OPG in the subgroup of women. These scatterplots have now been addded to the manuscript (Figure 2).

3. Section 2.3 “When we analyzed separately the group of women from that of men, the correlation between testosterone and proinflammatory cytokines could no longer be found in any group.” Does this statement mean that the associations were no longer significant or that the direction or magnitude of the associations changed? Please clarify.

Reply: what we meant was that the association was no longer significant/was not present any more. This has been clarified on line 106.

4. It would be good to add analysis as in table 2 separately by sex. Even if associations are not significant it is useful to assess whether there is any heterogeneity in associations between males and females.

Reply: Taking into account the Reviewer suggestion, we have added 2 new tables, which are Table 3 and Table 4, where we report the correlation separately by sex.

5. Have the assays used been assessed for reproducibility?

Reply: The assays have been assessed for reproducibility as specified on lines 254-258.

6. I suggest to use the word ‘parameters’ only for statistical parameters to be estimated and use ‘variables’ or something else for quantities that are measured.

Reply: The word “parameters” has been changed to “variables” throughout the text.

7. I suggest instead of ‘the significant level was set at 5%’ to say ‘p-values less than 5% were considered as evidence of an association’ or something similar.

Reply: We have changed the expression “‘the significant level was set at 5%” to “p-values less than 5% were considered as evidence of an association” on line 269.

Reviewer 2 Report

The authors, here present the difference in circulating pro-inflammatory cytokines levels in a cohort of healthy subjects mainly divided for their gender difference.

The overall approach is quite simple and few parameters are taken into account to reach the authors' conclusions.

Several scientific papers already showed the difference between male and female regarding their cytokine profile and its potential role in inflammatory-based pathologies: PMID:16153437; PMID: 31230049; PMID:27546235. As correctly reported by authors in the number 7 of the bibliography, similar results were already described by Bouman et al. In the same article, indeed, the role of testosterone was also discussed and its role upon IL1 and TNFalpha production was hypothesized as well. However, such effects are still debated and further investigations are needed (PMID: 29925283). Authors should have probably speculated more about Testosterone role since its variation is here described as the most correlated one to pro-inflammatory cytokines increasing. Moreover, considering the title of the work proposed, a comparison between the  pro-inflammatory cytokines levels in healthy adult versus patients diagnosed with atherosclerosis ,or versus subjects with a family medical history with higher chance to develop atherosclerosis than the one proposed, would have been advantageous.

Overall and in my opinion, in light of the existence of the over-mentioned literature and being the pathophysiology of atherosclerosis notoriously driven by pro-inflammatory and anti-inflammatory imbalance (PMID: 29925283), the work proposed, lacks of novelty and it should undergo extensive revisions and deeper implementations for its publication

Author Response

We thank the Reviewer fot the valuable feedback which has helped us improve the quality of the manuscript. Please find below our point-by-point reply. The changes have been highlighted by blue ink throughout the text.

Several scientific papers already showed the difference between male and female regarding their cytokine profile and its potential role in inflammatory-based pathologies: PMID:16153437; PMID: 31230049; PMID:27546235. As correctly reported by authors in the number 7 of the bibliography, similar results were already described by Bouman et al. In the same article, indeed, the role of testosterone was also discussed and its role upon IL1 and TNFalpha production was hypothesized as well. However, such effects are still debated and further investigations are needed (PMID: 29925283). Authors should have probably speculated more about Testosterone role since its variation is here described as the most correlated one to pro-inflammatory cytokines increasing. Moreover, considering the title of the work proposed, a comparison between the pro-inflammatory cytokines levels in healthy adult versus patients diagnosed with atherosclerosis ,or versus subjects with a family medical history with higher chance to develop atherosclerosis than the one proposed, would have been advantageous.

Reply: Taking into account the Reviewer suggestion, new references as well as the references indicated have been added to the manuscript, whose discussion has been extensively revised. Given that our results do not indicate that there is an independent association between testosterone and the proinflammatory cytokines, we have tried to clarify this point and the conflicting results that are present in the literature, showing that sex is associated with proinflammatory cytokines, but not testosterone or gonadal hormones.   

Overall and in my opinion, in light of the existence of the over-mentioned literature and being the pathophysiology of atherosclerosis notoriously driven by pro-inflammatory and anti-inflammatory imbalance (PMID: 29925283), the work proposed, lacks of novelty and it should undergo extensive revisions and deeper implementations for its publication.

Reply: To improve the quality of the manuscript we measured two new circulating meadiators that have been linked to atherosclerosis and are connected to those that we measured, which are RANKL and ACE2, which do not change between women and men. Interestingly, given that ACE2 seems to be the receptor for the coronavirus, our results suggest that the different vulnerability between women and men to the SARS-Cov2 could be ascribed to proinflammatory cytokines (such as IL-6), rather than to ACE2 (lines 211-219).

Round 2

Reviewer 1 Report

The authors have address previous comments.

Reviewer 2 Report

The authors has undoubtedly increased the value of their manuscript by improving their analysis and the text.

As previously underlined, no comparison between healthy adults and patients diagnosed with atheroslerosis ,or between healthy subjects and adults with a family medical history of atherosclerosis was proposed. In my opinion, this should be the main comparison to be performed if authors want to refer to the factors analysed as pro-atherogenic besides as pro-inflammatory.

However, as being a descriptive work and being such issues only conceptual, the general improvement was appreciated so I positively consider this work for its publication